# Learning Motion Refinement for Unsupervised Face Animation

**Jiale Tao**[1][*]   **Shuhang Gu**[1]   **Wen Li**[2][†]   **Lixin Duan**[1,2]
School of Computer Science and Engineering, UESTC[1]
Shenzhen Institute for Advanced Study, UESTC[2]
{jialetao.std, shuhanggu, liwenbnu, lxduan}@gmail.com

## Abstract

Unsupervised face animation aims to generate a human face video based on the appearance of a source image, mimicking the motion from a driving video. Existing methods typically adopted a prior-based motion model (*e.g.*, the local affine motion model or the local thin-plate-spline motion model). While it is able to capture the coarse facial motion, artifacts can often be observed around the tiny motion in local areas (*e.g.*, lips and eyes), due to the limited ability of these methods to model the finer facial motions. In this work, we design a new unsupervised face animation approach to learn simultaneously the coarse and finer motions. In particular, while exploiting the local affine motion model to learn the global coarse facial motion, we design a novel motion refinement module to compensate for the local affine motion model for modeling finer face motions in local areas. The motion refinement is learned from the dense correlation between the source and driving images. Specifically, we first construct a structure correlation volume based on the keypoint features of the source and driving images. Then, we train a model to generate the tiny facial motions iteratively from low to high resolution. The learned motion refinements are combined with the coarse motion to generate the new image. Extensive experiments on widely used benchmarks demonstrate that our method achieves the best results among state-of-the-art baselines.

## 1   Introduction

Face animation aims to generate a human face video based on the appearance of a source image to mimic the motion from a driving video. It has gained increasing attention from the community in recent years, due to its great potential in various applications including face swapping, digital humans, video conferencing, and more. Additionally, the emergence of large-scale face video datasets [18, 44] has further contributed to this growing interest.

Existing works on face animation can generally be divided into two categories: model-based and model-free methods (a.k.a supervised and unsupervised methods). On one hand, model-based methods [7, 33] typically use predefined 2D or 3D facial priors, such as 2D landmarks [20] and 3DMM parameters [2], to provide pose and expression information in the generated videos. While these methods have the advantage of describing accurate face poses, they are limited by overly strict structure priors that are not robust to hair and neck motions, which can heavily affect their performance in real-world applications. On the other hand, model-free animation is a more challenging and interesting problem, which aims to automatically learn the motion patterns without using predefined 2D or 3D facial models, thus being more flexible and robust in real-world scenarios. With the emergence of large-scale face video datasets, model-free methods have become more popular in

---

[*]Codes will be available at https://github.com/JialeTao/MRFA/
[†]The Corresponding Author



Figure 1: Illustration of the non-prior based motion refinement. Initially, a prior motion such as an affine transformation defines a motion flow. However, after the refinement, this flow can be matched to a new deformation that more realistically models underlying face deformations.

recent years [24, 25]. Among model-free methods, a local prior motion model is usually assumed to transform the sparse corresponding keypoints to the dense motion flow between the source and driving image. More recently, researchers have explored combining the two types of methods [40], such as using prior motion models to learn motion flow for warping source features while simultaneously employing 3DMM parameters to refine and enhance warped features.

In this work, we focus on the model-free face animation task, considering its flexibility for practical applications. Existing model-free face animation methods typically operate under the assumption of a local prior motion model, that is to assume the local motion between a source and driving image generally follows a parametric model such as affine transformation. For example, Siarohin *et al*. [24] proposed a local affine motion model, Siarohin *et al*. [25] and Tao *et al*. [29, 30] further constrained the learning process of the affine matrix, while others proposed a local thin-plate-spline motion model [43]. While these methods have the advantage of facilitating the process of learning motion and the ability to learn significant keypoints, their limitations are also obvious. Firstly, assuming that the human face is locally rigid may not be appropriate, resulting in sub-optimal motion models that fail to capture tiny motions in local facial areas. Secondly, accurately predicting geometric parameters such as the affine matrix from a single image is not easy. To address these issues, we propose a non-prior-based motion refinement approach to compensate for the inadequacy of existing prior-based motion models as illustrated in Fig. 1. Specifically, inspired by recent advances in optical flow [12, 28, 31], we utilize the keypoint features in building a correlation volume between the source and driving images, which represents the structure correspondence between the two images across all spatial locations. This correlation volume serves as non-prior motion evidence, according to which we employ a model to iteratively generate finer motions that refine the coarse motion predicted by prior motion models. In summary, compared to existing prior-based motion models, our approach exhibits a more powerful motion representation in terms of learning finer motions and thus is capable of achieving more realistic face animations.

We conduct extensive experiments on challenging face video benchmarks, and the superior performance of our method over state-of-the-art baselines highlights that learning motion refinement is advantageous for enhancing the existing prior-based motion models on face animation.

## 2    Related work

**Face animation:** Recent works on face animation can be roughly categorized into model-based and model-free methods, and more recently some works [40] tried to combine the two types of methods.

Model-based methods [4, 6, 7, 15, 21, 32, 36, 37, 38, 39] leverage predefined facial prior, such as 2D landmarks and 3D face models, to guide the pose or expression of the generated output. For instance, FSGAN [20] used facial landmarks as input to a generator directly to reenact a source face. Similarly, SAFA [33] used a 3D morphable model to aid in expression transfer. These methods are advantageous in modeling accurate face poses but suffer from the too-strict structure prior, which is not robust to neck and hair motions that are prevalent in practical scenarios.

In contrast, model-free methods [9, 10, 23, 24, 25, 30, 35, 42] tend to have better generalization performance and have become more popular due to the advent of large-scale face video datasets. These methods assume a local prior motion between subjects and learn local keypoints and corresponding motion transformation parameters in an unsupervised manner from videos. FOMM [24] is a pioneer work that proposed a local affine transformation model for image animation. Subsequently, MRAA [25], DAM [30], and MTIA [29] constrained the affine motion model learning process with stronger prior, such as PCA decomposition and cooperative motions. In fact, these methods assume a locally rigid object and mainly apply to human bodies, having limited performance on

human faces due to the easily violated locally rigid assumption. TPSM [42] proposed to adopt local thin-plate-spline motions for general animations, while DaGAN [10] used face depth information to better learn the keypoints and affine matrices. However, these methods still fall into the paradigm of prior-based motion models, which have difficulty in modeling tiny motions in local face areas. There are also some works [17, 34] that extremely abandoned the prior-based motion representation, they either used 3D head rotation supervision or utilized multiple source images for compensating the loss of motion prior. In this paper, we try to push the limit of single-image-based unsupervised face animation and propose a non-prior-based motion refinement approach, which overcomes the inadequacy of existing prior-based motion models by learning the finer motions.

**Optical flow:** Optical flow estimation has been a longstanding research problem in computer vision [11, 12, 27, 28]. Its objective is to estimate the motion between consecutive video frames. It's worth noting that the key part of unsupervised facial animation is also to estimate the motion flow between two images of the face. The main difference between the two tasks lies in the fact that optical flow deals with frames depicting the same scene, whereas facial animation focuses on transferring the motion between video frames featuring different subjects. This fundamental difference leads to variations in the methods employed for the two tasks. In more detail, optical flow computes the visual feature similarities between two frames. On the other hand, facial animation aims to disentangle the structural and visual appearance features and uses the structural feature to establish motion representations. Our method is inspired by the optical flow technique named RAFT [31]. Its fundamental idea is to develop a 4D correlation volume based on the visual features of two frames and use it to iteratively refine the motion flow. Similar to this concept, we build a structural correlation volume in face animation based on keypoint features; moreover, we introduce a multi-scale refinement mechanism to maximally leverage the source features, which helps to capture finer face motions.

## 3    Method

Our method consists of four key components: (1) a coarse motion estimation module based on a prior motion model, (2) a structure correlation volume calculator, (3) a motion flow updater, and (4) an image generator. While the first and last components are similar to existing face animation methods [24, 25, 29, 30], the second and third components form our proposed non-prior-based motion refinement module. The overview of our method is presented in Fig. 2. Similar to existing methods [24, 25, 29, 30] for achieving face animation, we first estimate the motion flow between the driving and source images. Then, we use the motion flow to warp the source features which is decoded by the image decoder, and generate the final animation video frame by frame. The difference is that in addition to estimating a coarse motion flow based on a prior motion model, we further estimate the motion refinement by our proposed motion refinement module. The coarse motion and the motion refinement are combined to warp source features. In the following subsections, we will first describe the existing prior-based motion models and then introduce our proposed non-prior-based motion refinement module.

### 3.1    Prior based Motion Estimation

Existing methods [24, 25, 29, 30, 40, 42] generally employ the affine prior and thin plate spline prior to model local facial motion. For the sake of simplicity, we adopt the local affine motion model as our formulation. This model assumes that a human face is composed of several individual parts and each part's motion is defined by an affine transformation. The parameters of these affine transformations are predicted by a keypoint detector, as shown in Fig. 2(a). The local affine transformations are then transformed into a set of part motion flows. A dense motion network predicts a composition mask, which is used to weigh the set of part motion flows to obtain a single motion flow, and an occlusion map which is used to mask the warped source feature in the occluded area.

Formally, we denote by $S$ and $D$ the source and driving images. A set of keypoints $\{p_s^k, p_d^k\} \in \mathbb{R}^{2\times1}$ and corresponding affine parameters $\{A_s^k, A_d^k\} \in \mathbb{R}^{2\times2}$ are predicted by the keypoint detector, where $k = 1, ..., K$. These affine parameters predicted from a single image are assumed to be the transformation to an abstract reference image [24]. Additionally, let $z \in \{0, 1, ..., h \times w - 1\}$ denote any pixel coordinate of the driving image, where $h$ and $w$ define the resolution of the motion flow which in this stage is set to 4 times downsampled image resolution. The part motion flow between

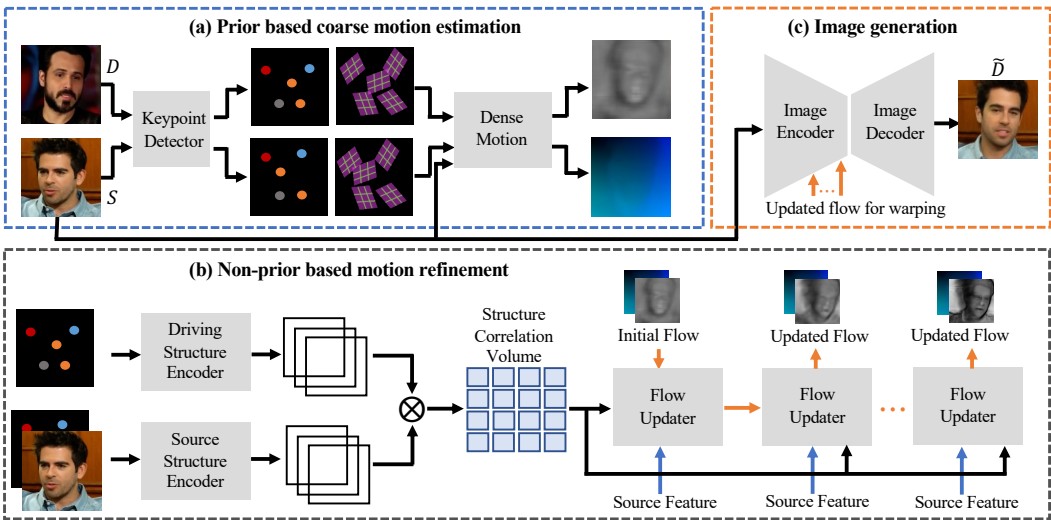

Figure 2: Overview of the pipeline. Our method consists of three modules: $(a)$ **Prior-based coarse motion estimation module** first estimates a coarse motion flow based on a prior motion model. $(b)$ Our proposed **non-prior based motion refinement module** constructs a 4D structure correlation volume that provides the non-prior motion evidence, based on which the coarse motion flow is iteratively refined. $(c)$ The **image generation module** for both encoding multi-scale source features and decoding multi-scale warped source features.

the driving and source images is derived using the following formula:

$$\mathcal{T}_{S \leftarrow D}^{k}(z) = p_s^k + A_s^k (A_d^k)^{-1} (z - p_d^k), \tag{1}$$

Intuitively, Equation (1) represents the flow format of an affine transformation that displays the coordinate correspondence between two images. In addition to the $K$ part motion flows, a motion flow of the background, which is typically considered to be static, is also represented by:

$$\mathcal{T}_{S \leftarrow D}^{0}(z) = z, \tag{2}$$

Using the predicted composition mask $M \in \mathbb{R}^{H \times W \times (K+1)}$, we can obtain the motion flow between the driving and source image as follows:

$$\mathcal{T}_{S \leftarrow D}(z) = \sum_{k=0}^{K} M^k(z) \cdot \mathcal{T}_{S \leftarrow D}^{k}(z), \tag{3}$$

We treat the Equation (3) as a coarse estimation of the motion flow, and in the following sections, we denote it using $\mathcal{F}_0 = \mathcal{T}_{S \leftarrow D}(z)$ without special indication. Additionally, we also denote $O_0$ as the occlusion map estimated by the prior motion model. Note that we derived $\mathcal{F}_0$ and $O_0$ by the affine prior, it is also suitable to use the thin plate spline prior for estimating the coarse motion flow.

### 3.2 Non-prior based Motion Refinement

Prior-based motion models have the advantage of facilitating learning motion in the early stages and can learn meaningful keypoints. However, the prior assumption such as a locally rigid face may not be appropriate, limiting them on learning finer facial motions. In this subsection, we present our non-prior-based motion refinement approach that compensates for the limitations of prior-based motion models.

Our motion refinement process is iterative, using the initial flow estimation obtained from a prior motion model, as described in the previous subsection. First, we construct a structure correlation volume to represent pixel-level structure correspondence between driving and source images. This volume remains fixed throughout later iterations. For each iteration, we use the current flow estimation to access the correlation volume and extract neighboring patch features. The cut correlation values, along with the warped source feature and current flow estimation, are sent to the flow updater to

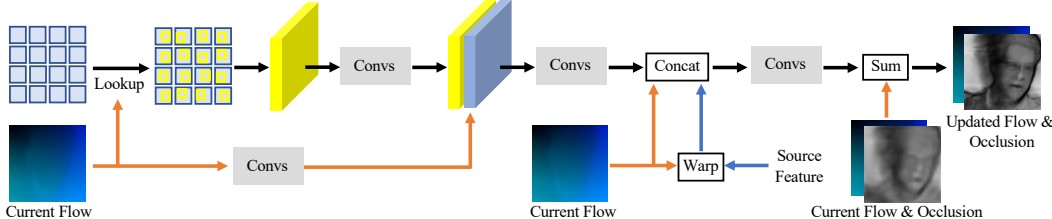

Figure 3: Details of the motion flow updater. It outputs the updated motion flow and occlusion map by processing three types of input features: warped source image feature, correlation feature extracted from the structure correlation volume using a lookup operation, and the current motion flow. The correlation feature and input motion flow are passed through two separate convolution modules, and their results are concatenated to yield an intermediate motion feature. This intermediate feature, together with the input motion flow and the warped source feature, is combined and input to a convolution module, which produces the residual flow and occlusion map.

obtain a refined flow. Note that the iteration proceeds from low to high resolutions, thereby enhancing the ability to capture finer facial movements. Fig. 3 provides a more detailed depiction of a single iteration of our motion flow update procedure. We will now introduce each component and operation.

**Structure correlation volume:** Inspired by recent advancements in optical flow [31], we propose a similar approach of constructing a structure correlation volume, which is considered to provide non-prior motion evidence for refining a coarse motion flow. We accomplish this by first encoding the source and driving structure features with two separate encoders. The driving structure encoder takes only the driving keypoints as input, while the source structure encoder concatenates the 4 times downsampled source image and keypoints as input. The source image is considered to provide dense visual context information for the sparse keypoints, which helps to densify the source structure. Keypoints are encoded with a fixed variance using Gaussian heatmaps centered on their positions. After encoding, the driving and source structure features are multiplied along the channel dimension to obtain a $4D$ structure correlation volume $C \in \mathbb{R}^{h \times w \times h \times w}$. The first two dimensions align with the driving coordinate system, while the last two represent the source coordinate system. Each indexed value in the first two dimensions represents the correspondence confidence between the driving pixel position and all source pixels. Note that the calculation of the structure correlation volume is carried out at a resolution that is 4 times lower than that of the original image. We also used a pyramid of correlations, which is similar to the optical flow approach [31]. The pyramid is created by pooling the structure correlation volume repeatedly with a series of scale factors. Further details can be found in the supplementary material.

**Correlation lookup:** For each iteration of the updates, we utilize the current input motion flow to lookup a $\{(2r + 1) \times (2r + 1)\}$-patch correspondence values centered at the flow position, where $r$ is the radius of the patch. This patch feature is further flattened to the channel dimension, resulting in a $h_i \times w_i \times (2r + 1)^2$ correlation feature $C_i$, where $h_i$ and $w_i$ are the flow resolutions of the $i$'th iteration. The lookup process is depicted on the left side of Fig. 3 and the correlation features are denoted by the yellow cuboid.

**Motion flow update:** The motion flow updater is comprised of four convolution modules that share parameters across all iterations. During each iteration, the flow updater takes in the current motion flow, extracted correlation values, and the warped source feature as inputs to produce outputs of updated motion flow and occlusion map.

To be more specific, we represent the current input motion flow as $\mathcal{F}_{i-1}^{\uparrow}$ which is upsampled twice from the previous iteration's output. The correlation feature is represented by $C_i$, and the source input feature encoded by the image encoder is represented by $f_i$. Firstly, the residual flow $d\mathcal{F}_i$ and the residual occlusion map $d\mathcal{O}_i$ are generated by the flow updater $\psi$:

$$d\mathcal{F}_i, d\mathcal{O}_i = \psi(\mathcal{F}_{i-1}^{\uparrow}, C_i, \mathcal{F}_{i-1}^{\uparrow}(f_i)), \tag{4}$$

where $\mathcal{F}_{i-1}^{\uparrow}()$ denotes the warping operation. The update process is then formulated as follows:

$$\mathcal{F}_i = \mathcal{F}_{i-1}^{\uparrow} + d\mathcal{F}_i, \mathcal{O}_i = \mathcal{O}_{i-1}^{\uparrow} + d\mathcal{O}_i. \tag{5}$$

We begin the iteration process at the resolution of the image downsampled by 32 times. The initial input motion flow for the first iteration is provided by:

$$\mathcal{F}_0^{\uparrow} = \text{Downsample}(\mathcal{F}_0, 8), \tag{6}$$

as $\mathcal{F}_0$ is computed at resolution $H/4 \times W/4$. The iteration is stopped once the maximal resolution iteration is accomplished, i.e., the image size. It should be noted that the initial flow input, $\mathcal{F}_0^{\uparrow}$, loses a significant amount of information due to the downsampling operation. To ensure preserving as much information from $\mathcal{F}_0$ as possible, we reformulate Equation (5) as:

$$\mathcal{F}_i = \text{Resize}(\mathcal{F}_0) + d\mathcal{F}_i + d\mathcal{F}_{i-1}^{\uparrow} + d\mathcal{F}_{i-2}^{\uparrow\uparrow} + ..., \tag{7}$$

where the initial flow $\mathcal{F}_0$ is directly resized to the current resolution. The similar rule is also applied to the occlusion map.

To summarize, the non-prior-based motion refinement module yields multi-scale refined motion flows and occlusion maps $\{\mathcal{F}_i, \mathcal{O}_i\}_{i=1}^N$, which are utilized in the process of image generation for warping source features.

**Non-prior-based initialization:** As previously discussed, we have been utilizing a prior motion model to estimate coarse motion flow owing to its effectiveness in facilitating learning motion. However, we have also proposed a non-prior-based initialization approach. Specifically, we reshape the structure correlation volume to $h \times w \times (h \times w)$ and denote the reshaped $C$ as $C^r$. We then perform a softmax operation on the final dimension, treating the result as an attention matrix. This matrix is subsequently employed to sum up an identity grid. The non-prior-based initialization is formulated as follows:

$$\mathcal{Q}_{S \leftarrow D}(z) = \sum_{i=1}^{h \times w} i \cdot \text{Softmax}(C^r)[z, i] \tag{8}$$

Where $z$ indexes the driving spatial dimensions and $i$ denotes the reshaped source dimension. In this way our proposed method becomes totally non-prior-based, we study this initialization in Section 4.

## 3.3  Image generation

As previously mentioned, we have prepared multi-scale motion flows, occlusion maps, and source features $\{\mathcal{F}_i, \mathcal{O}_i, f_i\}_{i=1}^N$. To better synthesize facial details, we use a Unet-like image generator which effectively combines the multi-scale source features with the refined multi-scale motion flows. In each layer of the image decoder, we integrate information from the current warping and previous generation by multiplying them with the occlusion map and reversed occlusion map. The generation process can be summarized as follows:

$$\text{out}_1 = \mathcal{F}_1(f_1) \cdot \mathcal{O}_1 \tag{9}$$

$$\text{out}_{i+1} = \mathcal{F}_{i+1}(f_{i+1}) \cdot \mathcal{O}_{i+1} + \text{UpBolck}(\text{ResBolck}(\text{out}_i)) \cdot (1 - \mathcal{O}_{i+1}), \tag{10}$$

where UpBolck defines a upsample-conv operation and ResBolck denotes a 2-layer resnet [8] block. At the last layer, the output is activated by a sigmoid function to obtain the generated image.

## 3.4  Training

Our method is trained end to end. Following previous methods [24, 25] and for the sake of simplicity, we only adopt the perceptual loss and the equivariance loss as our objective functions.

**Perceptual loss:** We adopt the multi-resolution perceptual loss [13] defined with a pre-trained VGG-19 [26] network. Given the driving image $D$ with resolution index $i$, the generated image $\tilde{D}$, and the feature extractor $\phi$ with layer index $l$, the perceptual loss can be written as follows:

$$\mathcal{L}_{per} = \sum_i \sum_l \left\| \phi_l(D_i) - \phi_l(\tilde{D}_i) \right\|_1. \tag{11}$$

**Equivariance loss:** Given a random geometric transformation $\mathbf{T}$ and a driving image $D$, the equivariance loss can be written as follows:

$$\mathcal{L}_{equi} = \sum_k \left\| \mathbf{T}(p_D^k) - p_{\mathbf{T}(D)}^k \right\|_1. \tag{12}$$

The overall loss is formulated by the sum of the two without weighting:

$$\mathcal{L} = \mathcal{L}_{per} + \mathcal{L}_{equi} \tag{13}$$

Table 1: Quantitative comparisons with state-of-the-art methods on the video self-reconstruction task. We present results on the Voxceleb1 and CelebV-HQ datasets. Our method generally achieves the best performance on all evaluation metrics.

| | Voxceleb1 | | | | | CelebV-HQ | | | | |
|---|---|---|---|---|---|---|---|---|---|---|
| | L1 | PSNR | LPIPS | AKD | AED | L1 | PSNR | LPIPS | AKD | AED |
| FOMM | 0.0412 | 23.85 | 0.171 | 1.284 | 0.135 | 0.0531 | 22.95 | 0.204 | 3.491 | 0.219 |
| MRAA | 0.0394 | 24.47 | 0.166 | 1.274 | 0.132 | 0.0433 | 24.65 | 0.173 | 1.852 | 0.167 |
| LIA | 0.0425 | 23.64 | 0.212 | 1.457 | 0.138 | 0.0507 | 22.95 | 0.235 | 1.982 | 0.179 |
| DAM | 0.0395 | 24.51 | 0.165 | 1.242 | 0.124 | 0.0496 | 23.58 | 0.189 | 1.899 | 0.180 |
| DaGAN | 0.0422 | 24.03 | 0.168 | 1.265 | 0.125 | 0.0652 | 21.37 | 0.249 | 8.704 | 0.307 |
| TPSM | 0.0396 | 24.84 | 0.160 | 1.203 | 0.122 | 0.0412 | 25.29 | 0.160 | 1.663 | 0.156 |
| MTIA | 0.0370 | 25.09 | 0.159 | 1.190 | 0.120 | 0.0405 | 25.63 | 0.157 | 1.532 | 0.153 |
| FNeVR | 0.0404 | 24.36 | 0.165 | 1.238 | 0.126 | - | - | - | - | - |
| Ours | **0.0354** | **25.57** | **0.151** | **1.155** | **0.108** | **0.0376** | **26.11** | **0.147** | **1.424** | **0.137** |

Table 2: Cross-identity evaluation on the Voxceleb1 dataset.

| | FOMM | MRAA | LIA | DAM | DaGAN | TPSM | MTIA | FNeVR | Ours |
|---|---|---|---|---|---|---|---|---|---|
| ARD | 3.122 | 2.678 | 3.883 | 2.669 | 3.090 | 2.724 | 2.794 | 2.755 | **2.399** |
| AUH | 0.850 | 0.729 | 0.772 | 0.717 | 0.751 | 0.668 | 0.676 | 0.691 | **0.647** |
| FID | 70.00 | 69.29 | 71.01 | 69.16 | 68.50 | 68.67 | 66.65 | 66.48 | **64.68** |

## 4   Experiments

**Implementation details:** We adopt two Hourglass networks [19] for the keypoint detector and the dense motion network, and a Unet [22] for the image generator similar to previous works [24, 25]. We further employ two hourglass networks for the driving and source structure encoders. The flow updater consists of four convolution blocks that share the parameters across the iterations. We train our method for 100 epochs on four NVIDIA A100 GPU cards or eight NVIDIA 3090 GPU cards. It takes about 24 hours for training. The number of keyoints is set to 10 following previous methods [24, 29, 30, 42]. The patch radius $r$ is set to 3 and the number of iterations is set to 6. The Adam optimizer [16] is adopted with $\beta_1 = 0.5$ and $\beta_2 = 0.999$, the initial learning rate is set as $2 \times 10^{-4}$ and dropped by a factor of 10 at the end of $60th$ and $90th$ epoch. We leave the architecture details in the supplementary material.

**Datasets:** We conduct experiments on the widely used Voxceleb1 [18] dataset and the recently collected more challenged CelebV-HQ dataset [44]. Voxceleb1 is a talking head dataset consisting of 20047 videos, among which 19522 are used for training and 525 are used for testing. The CelebV-HQ dataset consists of 35666 video clips, we randomly choose 500 of them for testing. All videos are resized to $256 \times 256$ for a fair comparison with existing methods.

**Metrics:** We measure the reconstruction quality using L1, Peak Signal-to-Noise Ratio (PSNR), and LPIPS [41] following [35, 40]. To evaluate the transferred motion quality we adopt Average Keypoint Distance (AKD) following previous methods [24, 25, 30, 42]. The identity quality of the generated videos is measured by Average Euclidean Distance (AED) [24, 25]. We use FID, ARD, and AUH metrics for evaluating cross-identity face animation following [5]. All these metrics are the lower the better except the PSNR.

**Baselines:** A bunch of baseline methods are compared, including FOMM [24], MRAA [25], LIA [35], DAM [30], DaGAN [10], TPSM [42], MTIA [29] and FNeVR [40]. All reported baseline results are obtained by evaluating the checkpoints from their Github repo. or retrain the provided codes, except that the FNeVR does not provide the training code and thus we do not report their results on the CelebV-HQ dataset. It is worth noting that most of these baseline methods adopt a prior motion model, while DaGAN [10] further explored the face depth information and FNeVR leverage the 3DMM parameters. For comparison with them, we adopt the motion transformer [29] for our prior motion estimation, while we also employ the affine motion model [24] and thin-plate-spline motion model [42] in the section of model ablations.

**Quantitative comparison:** Table 1 displays the same-identity reconstruction results, which is the standard evaluation setting used in previous methods [10, 24, 25, 29, 30, 40, 42] that reconstruct a driving video based on its first-frame appearance and all-frame motion. As can be seen, our

Table 3: Model ablations on the proposed non-prior based motion refinement module. We present results on the Voxceleb1 dataset. Baseline methods are all based on our implementation, which differentiates from original methods with a multi-scale feature fusion mechanism as described in Equation (10).

|  | L1 | PSNR | LPIPS | AKD | AED |
|---|---|---|---|---|---|
| NPMR | 0.0393 | 24.41 | 0.164 | 1.328 | 0.133 |
| FOMM | 0.0386 | 24.62 | 0.164 | 1.254 | 0.124 |
| FOMM + NPMR | 0.0367 | 25.15 | 0.156 | 1.224 | 0.114 |
| TPSM | 0.0370 | 25.09 | 0.159 | 1.231 | 0.120 |
| TPSM + NPMR | **0.0353** | 25.51 | 0.152 | 1.176 | **0.107** |
| MTIA | 0.0370 | 25.09 | 0.159 | 1.190 | 0.120 |
| MTIA + NPMR | 0.0354 | **25.57** | **0.151** | **1.155** | 0.108 |

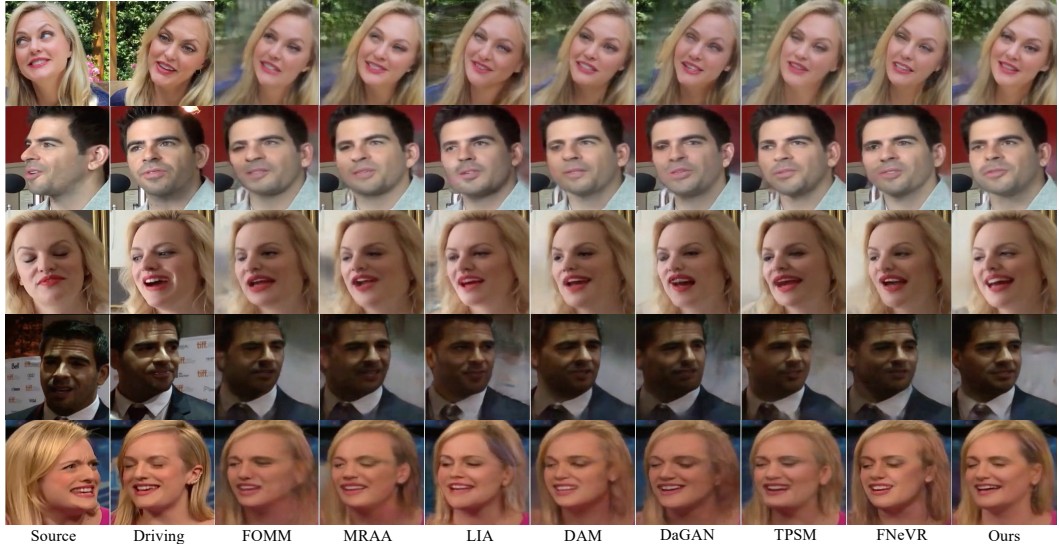

Figure 4: Same-identity video reconstruction on the Voxceleb1 dataset. Our method generally performs the best among a variety of baselines.

method outperforms other methods across all metrics on the two datasets. Specifically, large-margin improvements on the reconstruction metrics (L1, PSNR, LPIPS) demonstrate the advantages of learning motion refinement. What's more, our method achieved the best motion quality (the best score on the AKD metric) particularly on the more challenging CelebV-HQ dataset, highlighting the powerful motion representation capabilities of our non-prior-based motion refinement approach. The superior generated motion quality of our method is further validated by the best ARD and AUH metric in cross-identity experiments in Table 2. The best AED score and FID score also indicate that our method preserves the source face identity well during motion transfer. These significant observations are further supported by the qualitative results presented in Fig. 4, Fig. 5, and Fig. 6.

**Qualitative comparison:** We conducted experiments on both the same-identity video reconstruction and cross-identity face reenactment similar to previous methods [10, 29, 43], showcasing our qualitative results in Fig. 4, Fig. 5, and Fig. 6. Our method consistently demonstrates greater robustness against large motions (the first row of Fig. 4), detailed face deformations (the third row of Fig. 5 and the fourth row of Fig. 6), occlusions (the second and last row of Fig. 4), and varying light conditions (the fourth row of the Fig. 4), etc. Additionally, our approach outperforms other methods in generating precise facial details such as the eyes and lips, while avoiding common artifacts. To sum up, our methods can effectively learn non-prior-based motion refinement, exhibiting a significant improvement over prior-based motion representation that is limited on learning finer facial motions.

**Model ablations:** We performed a model ablation study to further evaluate the proposed **N**on-**P**rior-based **M**otion **R**efinement module. As previously discussed in Equation (8), in the non-prior-based initialization process, we initialize the motion flow using the structure correlation volume itself. We

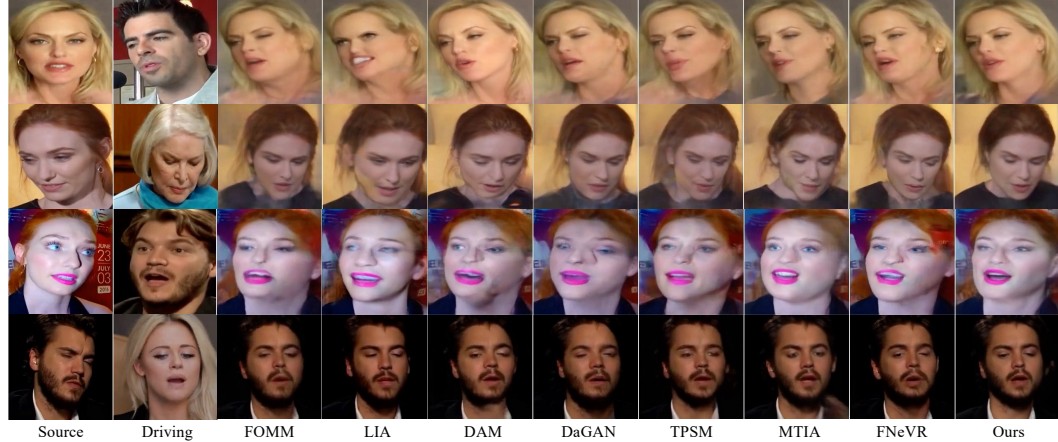

| Source | Driving | FOMM | LIA | DAM | DaGAN | TPSM | MTIA | FNeVR | Ours |

Figure 5: Cross-identity face reenactment on the Voxceleb1 dataset, our method consistently demonstrates superior performance in generating high-fidelity results compared to existing methods.

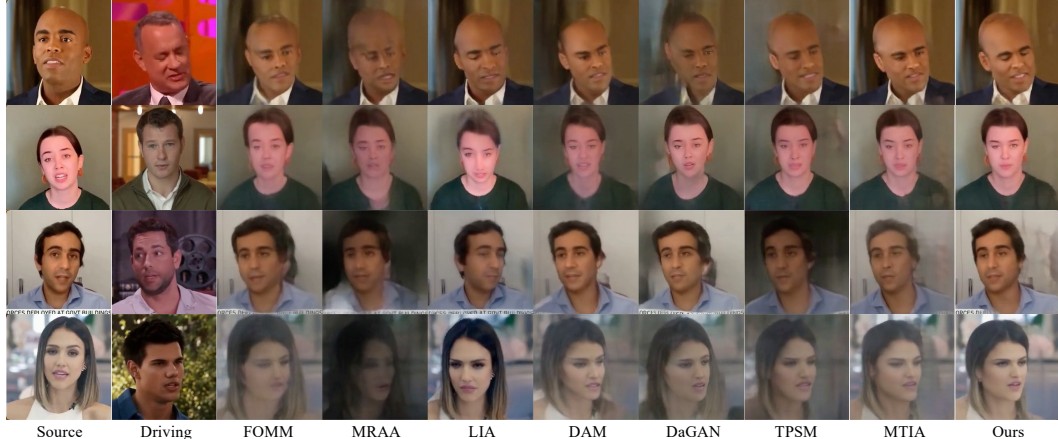

| Source | Driving | FOMM | MRAA | LIA | DAM | DaGAN | TPSM | MTIA | Ours |

Figure 6: Cross-identity face reenactment on the CelebV-HQ dataset, our method consistently demonstrates superior performance in generating high-fidelity results compared to existing methods.

denoted this model variant as NPMR. As shown in Table 3, our NPMR-only model did not yield good results, highlighting the importance of prior-based initial motion estimation for learning finer facial motions and the crucial role it plays in our non-prior-based motion refinement module. When combined with specific prior motion estimations, such as that from the affine and thin plate spline motion models, our method showed promising improvements over the original prior motion models. These combinations with different prior motion models significantly demonstrate the flexibility of the proposed non-prior-based motion refinement module, and its strong ability to promote the enhanced motion representation over existing prior motion models.

**Component ablations:** We conducted a thorough examination of the inputs and outputs of our non-prior-based motion refinement module through detailed ablations. Specifically, we explored the impacts of the model variants of the source structure encoder without the input of a source image, the flow updater without the input of warped source features, and the flow updater without the output of occlusion or flow (i.e. only updating either occlusion or flow and keeping the other the same as in the initialization process). The results are presented in Table 4. It is noteworthy that both the warped source feature and the source image play an important role in the motion refinement process as demonstrated by the decreased AKD and AED metric values on the $w/o$ warped source feature and $w/o$ source image variants. Importantly, if we solely refine the occlusion map without updating the motion flow (and vice versa), a significant decrease in AKD and AED metrics occurs, validating the significance of our motivation to refine the coarse motion flow.

Table 4: Component ablations of the inputs and outputs of the proposed non-prior-based motion refinement module. We present results on the Voxceleb1 dataset.

|  | L1 | PSNR | LPIPS | AKD | AED |
|---|---|---|---|---|---|
| $w/o$ flow | 0.0362 | 25.48 | 0.155 | 1.172 | 0.112 |
| $w/o$ occlusion | 0.0356 | 25.48 | 0.152 | 1.175 | 0.112 |
| $w/o$ warped source feature | 0.0357 | 25.52 | 0.153 | 1.160 | 0.110 |
| $w/o$ source image | 0.0354 | 25.45 | 0.152 | 1.165 | 0.113 |
| Ours full | **0.0354** | **25.57** | **0.151** | **1.155** | **0.108** |

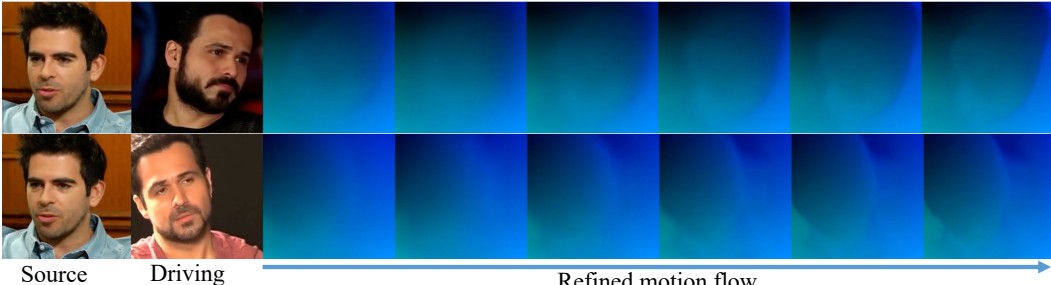

Figure 7: Motion flow visualization of the refining process, we show refined motion flows of all iterations. Note that flows of different resolutions are resized to the image resolution.

**Motion flow visualizations:** To further understand the motion refinement process, we visualize the refined motion flows in different iterations. As can be seen in Fig. 7, the motion flow gets refined from the initial iteration to the final iterations, corresponding to our description in Equation (7).

## 5   Conclusion

In this paper, we investigate the task of unsupervised face animation and propose a novel motion refinement approach to address the inadequacies of existing prior-based motion models for estimating finer facial motions. Our approach utilizes a structure correlation volume constructed from keypoint features, to provide non-prior-based motion information, which is used to iteratively refine the coarse motion flow estimated by a prior motion model. We conducted extensive experiments on challenging benchmarks, and our results demonstrate that our approach enhances the capability of prior-based motion representation through learning motion refinement.

**Limitations:** A major limitation of our approach is that it relies on the quality of the learned keypoints, as we mainly construct our proposed structure correlation volume using these keypoints. While current prior motion models may support the learning of significant keypoints, our method may still be limited by the quality of keypoints that are learned in an unsupervised fashion. To overcome this limitation, one potential solution would be to integrate model-based techniques that leverage facial priors, such as predefined keypoints that are more structured. Another limitation is the identity shift problem in cross-identity face animation, as keypoints often leak the face shape information of a driving video. The issue can be alleviated by employing relative motion transfer as in existing unsupervised animation methods [24, 25, 29]. Moreover, we could also explore prior motion models that can disentangle the global head motion and micro-expression motion, like facevid2vid [34].

**Social impact:** Our method has the potential to be used in creating deepfakes, which could have negative impacts. Therefore, individuals intending to use our technique for creating deepfakes should obtain authorization to use the respective human face images. Nevertheless, our method can also be utilized in creating imaginative image animations for entertainment purposes.

**Acknowledgement:** This work is supported by the National Natural Science Foundation of China (No. 62176047), Shenzhen Fundamental Research Program (No. JCYJ20220530164812027), and the Fundamental Research Funds for the Central Universities Grant (No. ZYGX2021YGLH208).

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

# Appendix

In this appendix, we present detailed information on the architectural design, evaluation metrics, correlation pyramid, and parameter analysis on the number of keypoints and the number of iterations.

**Architecture details:** Our system mainly consists of a keypoint detector, a dense motion module, the source and driving structure encoders, a flow updater, and an image generator. The keypoint detector and dense motion modules are implemented using blocks similar to those used in previous works [23, 29]. In contrast, we provide architecture details of our source and driving structure encoders and image generator in Figure 8. All the modules are included in our provided codes for easy reference and implementation.

**Evaluation metrics:** We mainly introduce four metrics that utilized third-party models for evaluation.

- Average Keypoint Distance (AKD [24]). This metric computes the average keypoint distance between generated and ground-truth images. It is designed to evaluate the pose quality of the generated images. We use existing detectors [3] to extract the facial landmarks.

- Average Euclidean Distance (AED [24]). This metric is designed to assess the identity quality of generated images based on specific feature representations, that are extracted from a pre-trained facial identification network [1]. The average Euclidean distance between generated and ground-truth video frames is computed.

- Average Rotation Distance (ARD [5]). We use the toolbox py-feat [14] to extract the Euler angles of the head poses, and then compute the average Euler angles distance between the generated and driving images. This metric evaluates the head pose quality.

- Action Units Hamming distance (AUH [5]). This metric measures the quality of facial expression, it computes the average Hamming distance between action units of generated and driving images. We use the toolbox py-feat [14] to extract facial action units.

**Details of the correlation pyramid:** We pool the structure correlation volume $C \in \mathcal{R}^{h \times w \times h \times w}$ in the last two dimensions to obtain the pyramidal structure correlation volume $\{C^i \in \mathcal{R}^{h \times w \times h/2^i \times w/2^i}\}_{i=0}^{P}$. In each iteration, we sample patch correlation features on all $C^i$'s in the pyramid, as the pyramid design aims to capture more rich motion features of different scales, which is inspired by the optical flow method RAFT [31]. Both the patch radius and the pyramid level can expand the search space of the structure correlation volume, resulting in the expanded correlation feature dimensions. In the main paper's experiments, we empirically set $P$ to 1 and $r$ to 3.

**Number of keypoints:** Following previous methods [24, 29], we configured the number of keypoints to 10 in the main paper. Here we conduct experiments to study this hyperparameter, with FOMM adopted as the prior motion model. As shown in Table 5, in a relatively sparse configuration such as 10-20 keypoints, our non-prior-based motion refinement approach can consistently improve the performance of the prior motion model.

**Number of iterations:** We set the iteration number according to the image resolutions in the main paper. Specifically, we start the iteration at a lower feature resolution $H/32 \times W/32$ that is meaningful for motion flow for warping, and end the iteration at the highest resolution $H \times W$, thus the total iterations are set to 6 for a 256-resolution image. By changing the highest resolution to $H/2 \times W/2, H/4 \times W/4, H/8 \times W/8$, we obtain iteration settings of $5, 4, 3$ accordingly. We then analyze the performance and run time of different iteration settings. As seen in the table, the FPS goes down as the iteration number increases. At the same time the performance is generally better with higher iterations, especially for the motion-related metric AKD and identity-related metric AED. Overall, the 5-iteration setting can be a good trade-off.

Table 5: Parameter analysis on the number of keypoints. We present results on the Voxceleb1 dataset.

|  | L1 | PSNR | LPIPS | AKD | AED |
|---|---|---|---|---|---|
| FOMM-kp10 | 0.0386 | 24.62 | 0.164 | 1.254 | 0.124 |
| NPMR+FOMM-kp10 | **0.0367** | **25.15** | **0.156** | **1.224** | **0.114** |
| FOMM-kp15 | 0.0376 | 24.91 | 0.161 | 1.245 | 0.123 |
| NPMR+FOMM-kp15 | **0.0360** | **25.34** | **0.154** | **1.199** | **0.108** |
| FOMM-kp20 | 0.0376 | 24.97 | 0.160 | 1.222 | 0.118 |
| NPMR+FOMM-kp20 | **0.0356** | **25.49** | **0.152** | **1.194** | **0.107** |

Table 6: Parameter analysis on the number of iterations. We present results on the Voxceleb1 dataset.

| Iteration Number | L1 | PSNR | LPIPS | AKD | AED | FPS |
|---|---|---|---|---|---|---|
| 3 | 0.0355 | 25.52 | 0.152 | 1.164 | 0.109 | **21.30** |
| 4 | **0.0353** | **25.57** | 0.151 | 1.156 | **0.107** | 20.45 |
| 5 | 0.0354 | **25.57** | **0.151** | **1.151** | 0.108 | 19.56 |
| 6 | 0.0354 | **25.57** | **0.151** | 1.155 | 0.108 | 18.57 |

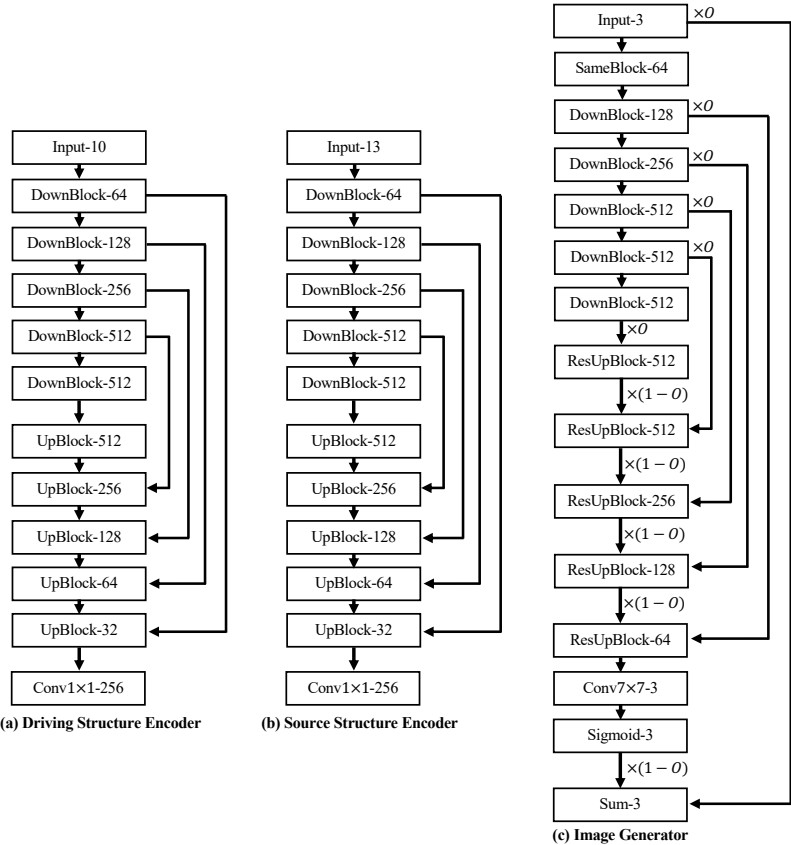

Figure 8: Detailed architectures of the driving structure encoder, the source structure encoder, and the image generator.

