# OpenReview forum: "Learning Motion Refinement for Unsupervised Face Animation"
_NeurIPS.cc/2023/Conference — NeurIPS 2023 poster_

### Official Review · Reviewer_zuqN · 2023-06-26

**Soundness:** 3 good
**Presentation:** 3 good
**Contribution:** 2 fair
**Rating:** 5
**Confidence:** 5

**Summary:**

This paper anaylizes the limitations of existing face animation methods in capturing the finer facial motions, and hence design a non-prior-based motion refinement approach to achieve finer face motion transfer in local areas. A correlation volume between the source and driving images is constructed as non-prior motion evidence, and a refinement module is introduced to generate the fine facial motions iteratively. Extensive experiments show the effectiveness of this paper.

**Strengths:**

1. The idea of involving a non-prior based motion refinement module is effective to capture the fine motions in local areas.
2. The manuscript is well written and clearly states the main idea and contribution in face animation.
3. This paper performs extensive experiments, and the results of the same-identity video reconstruction task outperform many previous methods.

**Weaknesses:**

1. This paper concentrates on the motion issue, but the video results of this paper are not impressive enough. No obvious improvements can be seen given the video results on the self reconstruction.
2. This paper does not show the quantitative results in terms of some usual metrics, i.e. CSIM and FID, on the cross-identity reenactment task. I wonder the performance of this work in preserving identity.
3. The introduction should be improved, as it only concludes limited contributions.

**Questions:**

1. Could the authors give more explations about the meanings of flow in the figures? I can not see the changes after updation. There are obvious improvements on the occlusion map, does the improvement in motions mainly come from the enhanced occlusion map.
2. Since Face vid2vid [1] shows a promissing performance on the cross-identity reenactment task, it is suggested to give a comparison with Face vid2vid. It is good to provide the video comparisons.

   [1] Ting-Chun Wang, Arun Mallya, and Ming-Yu Liu. One-shot free-view neural talking-head synthesis for video conferencing. In CVPR, 2021.
1. It is better to provide computation cost evaluation at inference, i.e. Flops, memory, and FPS, compared to other SOTA methods.

**Limitations:**

This paper is effective to refine the motions to obtain better results, but is still limited to the quality of the learned keypoints, as the limitaions in the paper stated, which I think is more urgent to solve for unsupervised face animation. And this paper did not give comprehensive evaluations on the performance in identity preserving, which is important for face animation in practical applications.
I'm not sure whether it is qualified for the conference.  If the authors could give reasonable response for the problems stated in **Weakness** and **Questions**, I would like to consider improving my rating.

---

> ### Author Rebuttal · Authors · 2023-08-09
>
> ## Response to Reviewer zuqN
>
> ### Q1: No obvious improvements can be seen given the video results on the self reconstruction.
>
> On one hand, we will rearrange the video results to form a nine-square presentation for better comparison. On the other hand, we suggest zooming in our Figure 4 in the main paper for detailed comparison, since our method produces finer motions in local areas such as lips and eyes. And we have also uploaded more video results on the CelebV-HQ dataset.
>
> ### Q2: This paper does not show the quantitative results in terms of some usual metrics, i.e. CSIM and FID, on the cross-identity reenactment task.
>
> Thanks for the good suggestion. It is indeed our carelessness. And we have compared this metric in the **common response**. While our method performs slightly better on the CSIM metric, it produces higher-quality motion transfer, as suggested by the better ARD and AUH metrics.
>
> ### Q3: The introduction should be improved, as it only concludes limited contributions.
>
> Thanks for the suggestion. We will revise it in the later version. But are there any more specific suggestions? Since "it only concludes limited contributions." is a little confusing to me.
>
> ### Q4: Could the authors give more explorations about the meanings of flow in the figures? Does the improvement in motions mainly come from the enhanced occlusion map.
>
> * Sorry for the confusion. The flows in the figure are actually estimated from a source and driving image pair, and they come from two different iterations. For better understanding, we provide the refined motion flow across all iterations in the video (the last few seconds).
> * The improvement in motions mainly comes from the enhanced motion, but not the occlusion map. As indicated by the component ablation studies in Table A2 of the supplementary material, removing flow updation undergoes a considerable performance decrease. However, the updated occlusion map is still helpful for understanding the flow updation process, since it reflects the non-warping area which is reversed to the motion flow that controls the warping area.
>
> ### Q5: Comparison with Face Vid2Vid
>
> Thanks for the suggestion. We quantitatively compare our method with Face vid2vid on the Voxceleb dataset. As seen in the table, our method outperforms it by a large margin. We also provided the link to the video comparison, which can be accessed from the Area Chair, please check in for details. It can be seen that similar to other methods, Face vid2vid also fails in capturing finer face motions, and our method generally produces better animation.
>
> |  | L1     | PSNR  | LPIPS | AKD   | AED   |
> |:---------------------:|:------:|:-----:|:-----:|:-----:|:-----:|
> | FaceVid2Vid           | 0.0444 | 23.40 | 0.175 | 1.405 | 0.138 |
> | ours                  | **0.0353** | **25.51** | **0.152** | **1.176** | **0.107** |
>
> ### Q6: It is better to provide computation cost evaluation at inference, i.e. Flops, memory, and FPS, compared to other SOTA methods.
>
> Thanks for the good suggestion. We have analyzed this in the **common response**.

---

> > ### Comment · Reviewer_zuqN · 2023-08-12
> >
> > Thank the authors for providing such a comprehensive reponse to all my questions.
> >
> > For limited contributions, this paper only focuses on refining the motion representation through a non-prior-based motion refinement approach, which also brings increased inference cost. It is more like a incremental method based on current models. The experiment results show the improvements in terms of motion representation over the proirs, so I raise the rating to 5. But I am still a bit conflicted about whether it is qualified for the conference.
> >
> > For Face vid2vid, it is known that Face vid2vid performs good at identity preservation but fails in motion modeling. From the provided videoes, we can also observe this point. But I am confused why the generated quality is so bad.

---

> > > ### Author Response · Authors · 2023-08-13
> > > **Post discussion**
> > >
> > > Thank you for improving the rating! We address your remained concerns in the following.
> > >
> > > ### Discussion 1: For limited contributions, this paper only focuses on refining the motion representation through a non-prior-based motion refinement approach, which also brings increased inference costs. It is more like an incremental method based on current models.
> > >
> > > * For the contribution, we would like to emphasize that existing face animation methods generally ignore the importance of non-prior based motion modeling. Since the existing face animation framework generally consists of the two stages of **motion estimation** and **image generation**, improving motion representation is meaningful and necessary for this task. And we proposed an enhanced motion representation called the non-prior based motion refinement, which is proven to be effective through extensive experiments. Moreover, due to the uniqueness of the non-prior based motion modeling, our method is generalizable to improve a bunch of existing prior-based animation methods such as FOMM[1], TPSM[2], MoTrans[4], etc. This generalization ability also reflects the importance of modeling non-prior motion for face animation, indicating that our method is not incremental but a novel approach that is complementary to existing methods.
> > >
> > > * We acknowledge that the non-prior based motion refinement will bring some extent increased inference cost. But we still achieve a good tradeoff between efficiency and effectiveness, as in the common response, our method is only 3.26 FPS slower than FNeVR[5], and 1.39 FPS slower than TPSM, while faster than DaGAN[3].  We should also note that currently in face animation, with not-so-bad inference speed, better performance may be preferred than better speed. For example, the suggested method Face Vid2Vid is actually the slowest one among existing methods, due to the heavily used 3D convolution for 3D landmark estimation and motion estimation, and it only achieves 5.9 inference FPS and 593.5 GFLOPs under our same testing of a single NVIDIA 3090.
> > >
> > > [1] First Order Motion Model for Image Animation, NeurIPS 2019.
> > >
> > > [2] Thin-plate Spline Motion Model for Image Animation, CVPR 2022.
> > >
> > > [3] Depth-aware Generative Adversarial Aetwork for Talking Head Video Generation, CVPR 2022
> > >
> > > [4] Motion Transformer for Unsupervised Image Animation, ECCV 2022.
> > >
> > > [5] FNeVR: Neural Volume Rendering for Face Animation, NeurIPS 2022.
> > >
> > >
> > > ### Discussion 2: Better identity preservation of Face Vid2Vid. Why the generated quality is so bad.
> > >
> > >
> > > * The ability of identity preservation is indeed a limitation of our method, and it also exists in existing unsupervised animation methods. It should also be noted that the Face Vid2Vid used 3D head pose supervision during training, which may lead their method to better disentangling the global head motion and the local expression motion, thus achieving better identity preservation. We will include this as a limitation in our later version, and in future work, introducing 3D supervision may help us alleviate this problem.
> > >
> > > * On one hand, as you suggested, the Face Vid2Vid performs not well in motion transfer, though preserves better identity. And without losing generality, we have shown the representative videos which may a little magnify their inability in finer motion modeling. On the other hand, since the authors of Face Vid2Vid didn't release their source code, we reproduce it based on the [unofficial code](https://github.com/zhanglonghao1992/One-Shot_Free-View_Neural_Talking_Head_Synthesis) following DaGAN and FNeVR, though we have checked that the main components did correspond to the original paper, the detailed implementations may still cause some performance gap. But we believe this may not be significant as the reproducer communicated with the authors of Face Vid2Vid during the implementation.

---

> > > > ### Comment · Reviewer_zuqN · 2023-08-18
> > > >
> > > > Thank you for your rebuttal. I have read it and will respond with points for further discussion.

---

> > > > > ### Author Response · Authors · 2023-08-19
> > > > >
> > > > > Thank you for the reply. We look forward to your valuable comments!

---

### Official Review · Reviewer_SbHP · 2023-07-03

**Soundness:** 3 good
**Presentation:** 4 excellent
**Contribution:** 3 good
**Rating:** 6
**Confidence:** 3

**Summary:**

This paper proposes a method for face animation via learning to refine a coarse motion field. The refinement is performed in a recurrent manner using the previous iteration's motion, occlusion map and structure correlation volume.  Both the qualitative and quantitative results show improvement over prior art.

**Strengths:**

1) Qualitative results show that the proposed method is clearly model finer facial movement much better than prior art, especially around the eye. FNeVR is very close but the proposed method has fewer uncanny artifacts.

2) Quantitatively as well, the proposed method outperforms prior art.

3) The paper is well written.

**Weaknesses:**

1) Across all the results, I notice a strong identity shift during animation. While this is common (and seemingly worse) in prior art as well, it is important to quantify. One way to measure this is to report the FaceIDLoss or L2-loss as the head is rotated 15, 30, 60 degrees from the original head-pose. I believe this is an important evaluation to include for the sake of completeness.

**Questions:**

N/A

**Limitations:**

1) An important limitation of this method is that it is sensitive to the scale of the face of the driving video. For example, around the 1:47 of the supplementary video, the face structure of the animated face deviates significantly from the source and is actually closer to that of the target. This is unavoidable due to the use of 2D key points in the method, in fact it is expected. The authors must include this as a limitation of their work.

---

> ### Author Rebuttal · Authors · 2023-08-09
>
> # Response to Reviewer SbHP
>
> ### Q1: Identity shift during animation, and FaceIDLoss
>
> Thanks for the good suggestion. We have compared the common face identity preservation metric CSIM in the **common response**, and our method performs slightly better on this metric while much better on the motion related metrics.
>
> ### Q2: Limitation on the face scale
>
> Thanks for the suggestion. Indeed this is a limitation of our method, and we will include this point in the later version.

---

> > ### Comment · Reviewer_SbHP · 2023-08-19
> > **Rebuttal Update**
> >
> > I would like to thank the authors for the rebuttal. After reading it and the other reviews, I am inclined to keep my rating. I would strongly encourage the authors to include the table from Q3 of the general response in the main paper.

---

> > > ### Author Response · Authors · 2023-08-20
> > >
> > > Thank you for the reply and for keeping the rating! Following your suggestion, we will include the table of cross-identity evaluation in the main paper.

---

### Official Review · Reviewer_pDRj · 2023-07-04

**Soundness:** 3 good
**Presentation:** 3 good
**Contribution:** 2 fair
**Rating:** 4
**Confidence:** 3

**Summary:**

This paper contributes to the field of unsupervised face animation by introducing a novel motion refinement method to overcome the limitations of existing prior-based motion models, especially in estimating detailed facial movements.

The paper's approach introduces a new method which uses a structure correlation volume built from keypoint features. This provides motion information that does not rely on prior data. This information is used to iteratively refine the coarse motion flow estimated by a previous motion model.

The authors have conducted numerous experiments on challenging benchmarks to test the effectiveness of their approach. The results indicate that this new method enhances the capability of prior-based motion representation by learning how to refine motion. This suggests that their approach can effectively increase the accuracy of unsupervised face animation tasks.

**Strengths:**

In this research, the authors present a new unsupervised face animation approach that concurrently learns both coarse (global) and finer (local) facial motions. Their method integrates a local affine motion model to learn the global, coarse facial motion and a novel motion refinement module to compensate for the local affine motion model's ability to model more detailed facial motions in local areas.

The motion refinement process is based on the dense correlation between the source and driving images. To achieve this, a structure correlation volume is first constructed using the keypoint features of the source and driving images. The authors then train a model to iteratively generate minor facial motions from low to high resolution.

The learned motion refinements are combined with the coarse motion to generate the new image. After performing extensive experiments on widely used benchmarks, the method was found to deliver the best results among existing state-of-the-art methods. The authors have also committed to making the source code for their method publicly available in the future.

**Weaknesses:**

1. The goal of this article is to learn fine facial motion, but using the VoxCeleb dataset may not be sufficient for this purpose. In my view, finer facial motion refers to the ability to reproduce wrinkles, eyeballs, and micro-expressions at high resolution, which may require higher resolution datasets.

2. As noted by the author, the reliability of keypoints estimation heavily influences the quality of finer facial motion. It is not clear whether the improvement of existing models comes from more accurate keypoints estimation or the proposed module.

3. This paper is very similar to RAFT from ECCV2020 in terms of insight and specifically for the module. A clearer comparison between the two works and their essential differences would be helpful.

4. It is important to note that fine facial motion may not equal to better image quality. Therefore, a more thorough evaluation of the estimated motion would be beneficial. Additionally, conducting this task on higher resolution facial images would provide more convincing results.

5. The author sets the iteration number to 8, which is an important hyperparameter. It would be useful to know why this setting was chosen and whether it significantly affects the model's runtime.


**Questions:**

1. This paper is very similar to RAFT from ECCV2020 in terms of insight and specifically for the module, which raises novelty concerns.

2. Related works [1] are not discussed and compared in this paper.

[1] Tao, Jiale, Biao Wang, Tiezheng Ge, Yuning Jiang, Wen Li, and Lixin Duan. "Motion Transformer for Unsupervised Image Animation." In European Conference on Computer Vision, pp. 702-719. Cham: Springer Nature Switzerland, 2022.

**Limitations:**

Both limitations and social impact are discussed in this paper.

---

> ### Author Rebuttal · Authors · 2023-08-09
>
> ## Response to Reviewer pDRj
>
> ### Q1:  VoxCeleb dataset may not be sufficient for evaluating finer facial motion and higher resolution dataset may be more appropriate.
>
> Thanks for the suggestion, but we may not agree that the VoxCeleb dataset may not be sufficient for evaluating finer facial motion, as our experiments in the main paper had proved this (Figure 4, Figure 5 and Table 1) and results from other methods generally failed in capturing finer face motions. We here also perform experiments on the 512-resolution CelebVHQ dataset. As seen in the table, our method can still improve existing SOTA methods on all metrics on the high resolution setting. Since it is time-consuming to train on the 512-resolution dataset, we are not able to reproduce other baselines at this time, and we will include them in a few days. Thanks for the suggestion again.
>
> |                       | L1  | PSNR | LPIPS | AKD | AED |
> |:---------------------:|:---:|:----:|:-----:|:---:|:---:|
> | TPSM                  | 0.0435 | 24.91  | 0.193   | 3.035 | 0.158 |
> | SCORR+TPSM            | **0.0418** | **25.24**  | **0.185**   | **2.911** | **0.149** |
>
>
> ### Q2: Whether the improvement of existing models comes from more accurate keypoints estimation or the proposed module
>
> Thanks for the good question. If I'm not mis-understanding, so the question is whether the improvements come from that the proposed module promotes estimating more accurate keypoints, or come from that the proposed module did learn finer motions. We explain this in the following.
>
> Since our structure correlation volume is built on keypoints, the gradient on the structure correlation volume will back-propagate to the keypoint detector, thus it may affect the learning process of keypoints. We validate this by a simple but well-designed experiment on the Voxceleb dataset, that is to stop the gradient flow from our structure encoders to the keypoint detector, by simply detach the keypoints from previous gradient graph before sending them to the structure encoders. The operation is denoted by Ours-$sg$ in the Table. As seen, this operation didn't affect the final performance at all, and the stop gradient operation even performs slightly better. So we believe that the improvements come from that our proposed module did learn finer motions.
> |                       | L1     | PSNR  | LPIPS | AKD   | AED   |
> |:---------------------:|:------:|:-----:|:-----:|:-----:|:-----:|
> | Ours                  | **0.0353**| 25.51 | 0.152 | 1.176 | 0.107 |
> | Ours-$sg$             | **0.0353** | **25.54** | **0.150** | **1.175** | **0.105** |
>
>
> ### Q3: A clearer comparison and essential differences between the RAFT and our method.
>
> We initially discussed this in the related work of the main paper, and we here make a more detailed and clearer discussion in the **common response**. We sincerely hope you could reconsider the novelty of our method in leaning non-prior based motion in face animation.
>
> ### Q4: Fine facial motion may not equal to better image quality. A more thorough evaluation of the estimated motion would be beneficial. Additionally, conducting this task on higher resolution facial images would provide more convincing results.
>
> Thanks for the thoughtful question. In general, the unsupervised face animation methods (both existing and our method) consist of two stages: **motion estimation** and **image generation**. One can investigate better image generation modules to improve the image quality, such as FNeVR which adopted the SPADE[1] module as their image generator and considerably improves the generation performance as seen in their ablation studies. But we think finding better motion estimation is more meaningful and urgent in this area. So we made our motivation fall into learning motion refinements and our experiments did validate this motivation. For evaluating the motion quality, we think the metric **AKD** in the self reconstruction task, and the **ARD** and **AUH** metrics in cross-identity experiments that we present in Table A1 of the supplementary material, are sufficient for the purpose. We also conduct experiments on the high-resolution dataset in Q1. We sincerely hope that you could reconsider this point.
>
> [1] Semantic Image Synthesis with Spatially-Adaptive Normalization, CVPR 2019.
>
> ### Q5: Why set the current iteration number.
>
> Thanks for the question. Due to the length limitation of characters, we addressed this similar concern in the **Q1 of Reviewer mUhK**, please refer to that.
>
>
>
> ### Q6: Comparison and discussion with existing method [1].
>
> Thanks for suggesting the paper. This method is also prior-based animation methods, which employs a vision transformer for better learning affine motion relationship. So our non-prior based motion refinement module would still be effective for improving it. We thus make a direct comparison with it on the Voxceleb dataset, by applying our motion refinement module on the top of the motion transformer. As seen, the performance even surpass our original result in the main paper, which reflects the robustness of our non-prior based motion refinement module to different prior-based motion models. We will discuss this in the later version.
>
> |  | L1     | PSNR  | LPIPS | AKD   | AED   |
> |:---------------------:|:------:|:-----:|:-----:|:-----:|:-----:|
> | MoTrans               | 0.0370 | 25.08 | 0.160 | 1.191 | 0.120 |
> | SCORR+MoTrans         | **0.0352** | **25.64** | **0.150** | **1.164** | **0.108** |
>
> [1] Motion Transformer for Unsupervised Image Animation, ECCV 2022.

---

> > ### Comment · Reviewer_pDRj · 2023-08-18
> >
> > Thanks for providing the detailed rebuttal. I have read the other five comments and the rebuttal. The visualization results in the provided video are still not in high quality from my view.  I am ok to accept this paper but at the Borderline level.

---

> > > ### Author Response · Authors · 2023-08-19
> > > **On Updating the Rating**
> > >
> > > Thank you for the reconsideration of our work. We have uploaded more video results of the CelebV-HQ dataset, and it can be clearly observed that our method produced much better results than others. Thank you for the reply. And could you please update your rating?

---

### Official Review · Reviewer_mUhK · 2023-07-05

**Soundness:** 3 good
**Presentation:** 3 good
**Contribution:** 2 fair
**Rating:** 5
**Confidence:** 2

**Summary:**

The paper presents an unsupervised method for learning to create continuous video animations of faces given a single input image and a driving video of the same or a different subject. The method relies on the optical flow between the driving and source images to warp the features of the source image, which are then passed to a generator to create the final output frames. The main contribution of the proposed method lies in the two-step formulation of the flow estimation, in which starting from a coarse prediction using a known prior-based method (Affine Transforms, Thin Plate Splines) it iteratively produces updates to reach a more fine-grained flow (and occlusion) estimation with more local details. The flow updates are learnt in an unsupervised way without using prior models, but by building a correlation volume between the source and driving images. The paper includes experiments, generated images and supplementary videos that validate the authors' claims for more detailed animations in comparison to similar methods.


**Strengths:**

- The paper is well written overall and it is easy to follow along with the presented concepts and results.
- The paper includes experimental results that validate the authors' claims, as well as ablations that offer a better understanding about model design choices.
- The method produces higher-quality generated images/videos than the compared methods for unsupervised learning of face animation.
- The idea of using motivation from RAFT to built a similar correlation structure for estimating motion flow for face animation is interesting.

**Weaknesses:**

- Some design choices are not empirically or experimentally justified in the paper. For example, why are specifically 6 iterations used to update the initial coarse estimations? Why does it make sense to sample from the same correlation structure for all iterations, while each iteration operates at a different image scale?
- Even though the method performs well compared to previous methods, there is still some evident identity shift in cross-subject experiments, meaning that the shape of the face slightly changes from the original to the one of the driving frame. This is evident in Figure 5 (for example 2nd row) as well as in the supplemental video. This effect makes sense because the estimated flow between different subjects possibly includes more deformation information, rather than only deformation because of expressions or rigid motion. Disentangling identity from other motions is a common weakness of cross-identity animation methods, which exists in this method, too.

**Questions:**

- In 3.4 it is mentioned that a perceptual and an equivariance loss are employed for training. Is the model trained with both same and different subjects for source and driving frames?
- How are C^{i} combined to get the final values from the correlation pyramid?
- What is the effect of the correlation pyramid to the memory requirements of the method? How does the memory requirements compare with the compared methods?

**Limitations:**

The authors have included a section discussing the method's limitations and possible negative societal impact. A mentioned limitation (the correlation matrix relying on learned keypoints) might also be the reason for changing the facial shape in cross-subject animations.

---

> ### Author Rebuttal · Authors · 2023-08-09
>
> ## Response to Reviewer mUhK
> ### Q1: Some design choices are not empirically or experimentally justified in the paper. For example, why are specifically 6 iterations used to update the initial coarse estimations!
> Thanks for the careful review and sorry for the confused understanding. We set the iteration number according to the image resolutions. Specifically, we start the iteration at a lower feature resolution $H/32\times W/32$ that is meaningful for motion flow for warping, and end the iteration at the highest resolution $H\times W$, thus the total iterations are set to 6 for a 256-resolution image. By changing the highest resolution to $H/2\times W/2, H/4\times W/4, H/8\times W/8$, we obtain iteration settings of $5,4,3$ accordingly. We then analyze the performance and run time of different iteration settings. As seen in the table, the FPS goes down as the iteration number increases, while the performance is generally better with higher iterations especially for the motion-related metric AKD and identity-related metric AED. Overall, the 5-iteration setting can be a good trade-off.
>
>
> |  | L1              | PSNR           | LPIPS          | AKD            | AED            | FLOPs (G)      | FPS            |
> |:--:|:------:|:-------:|:-----:|:-----:|:-------:|:------:|:-------:|
> | 3                     | 0.0357          | 25.37          | 0.153          | 1.194          | 0.112          | **89.86** | **21.30** |
> | 4                     | 0.0354          | **25.54** | 0.152          | 1.188          | 0.113          | 91.20          | 20.45          |
> | 5                     | **0.0353** | **25.54** | **0.151** | 1.177          | 0.108          | 96.34          | 19.56          |
> | 6                     | **0.0353** | 25.51          | 0.152          | **1.176** | **0.107** | 116.5          | 18.57          |
>
>
> ### Q2: The identity shift in cross-subject experiment.
> Thanks for pointing out this. We addressed this concern in the **common response**. Though our method performs slightly better on the identity-preservation metric, it is indeed a limitation of our method, and in the future, we would like to explore leveraging 3D face information for helping alleviate this problem.
>
> ### Q3: Why does it make sense to sample from the same correlation structure for all iterations, while each iteration operates at a different image scale?
> Thanks for the detailed question. The correlation matrix is of size $h\times w\times h\times w$, where $h=H/4, w=W/4$, $H,W$ define the image size. The indexes of the first two dimensions are corresponded to the driving frame coordinate while that of the last two dimensions denote the source coordinate. By a simple downsampling or upsampling operation on the first two dimensions, we obtain the different-scale correlation matrixes which are used in different iterations. The reason why it makes sense to sample from the same correlation structure for all iterations is that, the warped source features at different resolution should share similar spatial structures, thus we can downsample or upsample the correlation matrix to the different resolution, and then use it to obtain refined motion flow at different resolution for warping different-scale source features.
> ### Q4: Is the model trained with both same and different subjects for source and driving frames
> No, similar to FOMM, the model is only trained on the same subjects for source and driving frames.
>
> ### Q5: How are $C^{i}$ combined to get the final values from the correlation pyramid? What is the effect of the correlation pyramid to the memory requirements of the method?
> * For each $C^{i}$, as described in the line 177-179 of the main paper, we lookup a $(2r+1)\times (2r+1)$ patch correlation feature. We then concatenate all correlation features in the pyramid, result in a $P\times (2r+1)^2$-channel correlation feature, where $P$ denotes the pyramid level.
>
> * Actually, the correlation pyramid brings few additional memory requirements, as the concatenated correlation feature will be send to a convolution (with kernel size equal to 1) to get a fixed 128-channel feature. That means, the only increased memory requirements come from the increased convolution parameters of size $(P-1)\times (2r+1)^2\times 128$. In practice, we set $P=2, r=3$, and the increased parameter size is equal to $6272=0.006$M, leading to few additional memory requirements.
>
> ### Q6: How does the memory requirements compare with the compared methods?
> Thanks for the question. We addressed this concern in the **common response**.

---

> ### Author Response · Authors · 2023-08-21
>
> Dear Reviewer mUhK:
>
> Thank you for the recognition of our work. We addressed your concerns in the rebuttal. Could you please share your post comments? We are happy to discuss and address any remained concerns.
>
> Authors of the submission.

---

> > ### Comment · Reviewer_mUhK · 2023-08-21
> >
> > Thank you for the detailed response. My questions about design choices and memory requirements have been resolved after the rebuttal. Having read the other reviewers' comments and because identity shift is still evident in the provided videos, I would like to keep my score for the paper as is.

---

> > > ### Author Response · Authors · 2023-08-21
> > >
> > > Thank you for the reply and for keeping the score! As suggested, while we focus on the non-prior based motion refinement, identity shift is a common issue in existing unsupervised methods. And in the future, we would like to explore supervision from 3D face information to help alleviate this problem.

---

### Official Review · Reviewer_2aeH · 2023-07-13

**Soundness:** 3 good
**Presentation:** 3 good
**Contribution:** 3 good
**Rating:** 5
**Confidence:** 5

**Summary:**

1. This work proposed a  new unsupervised face animation approach to learn simultaneously the coarse and finer motions.
2. The results outperformed the state-of-the-art methods on two representative datasets.

**Strengths:**

1. The target task is important.
2. The method is intuitive and reasonable.
3. The results are good.
4. The paper is well-written.

**Weaknesses:**

1. Compared to prior model. The proposed non-prior models seems like a non-linear version of prior model based on affine transformation. When the number of keypoints increasing and the grid of affine transformation being much finer, will it approach the proposed non-prior model? Another related question is that why the proposed model works better than a local thin-plate-spline motion model (such as [38])?

2. Visualization. 1) In the visualization of comparison in video, the results of the proposed model looks close to FNeVR. Authors could also provide user study in visual quality for comparison. 2) Also, the results on CelebV-HQ is really important, since it is a much challenging dataset. Visualization results on this dataset is necessary, in comparison, ablation study and video. 3) Why didn't report the results of FNeVR on CelebV-HQ on Table 1?

I am inclined to accept this paper, if the concerns can be solved.

**Questions:**

None.

**Limitations:**

Yes, limitations are discussed in the main paper.

---

> ### Author Rebuttal · Authors · 2023-08-09
>
>
> ## Response to Reviewer 2aeH
> ### Q1: Will the increasing number of keypoints of the local affine motion model approach the non-prior based motion model? And why the proposed model works better than a local thin-plate-spline motion model?
>
> * Thanks for the good question. If one can learn the dense human face keypoints that are meaningful and structured, the local affine motion model will approach the non-prior based motion, as many local linear models will approach a good non-linear model. However, it is very hard to learn dense structured keypoints in unsupervised face animation. We conduct an experiment that configure the FOMM with 100 keypoints, the result in the table is denoted as "FOMM-kp100". It should be noted that the motion related metric AKD even goes through a decrease compared to 10 keypoint configuration. We found that many keypoints are non-meaningful and overlapped with other keypoints or located in the background. On the other hand, in a relatively sparse configuration such as 10-20 keypoints, our non-prior based motion refinement approach can consistently improve the performance.
> * The reasons why the proposed model works better than a local thin-plate-spline motion model are two folds. On one hand, though nonlinear, the local thin-plate-spline motion model is still a parametric (prior-based) motion model, thus the non-prior based motion still has the stronger ability to model complex deformations. On the other hand, in TPSM[37], it needs 5 geometric consistent keypoints to determine a local thin-plate-spline transformation, and 10 local transformations require 50 geometric consistent keypoints which heavily increases the learning difficulty. Though it used the dropout strategy in learning transformations, the learning difficulty may still limit their method in reaching the representation ability of local thin-plate-spline transformations.
>
> |  | L1     | PSNR  | LPIPS | AKD   | AED   |
> |:---------------------:|:------:|:-----:|:-----:|:-----:|:-----:|
> | FOMM-kp100            | 0.0371 | 25.15 | 0.162 | 1.288 | 0.116 |
> | FOMM-kp10             | 0.0386 | 24.62 | 0.164 | 1.254 | 0.124 |
> | SCORR+FOMM-kp10       | **0.0367** | **25.15** | **0.156** | **1.224** | **0.114** |
> | FOMM-kp15             | 0.0376 | 24.91 | 0.161 | 1.245 | 0.123 |
> | SCORR+FOMM-kp15       | **0.0360** | **25.34** | **0.154** | **1.199**| **0.108** |
> | FOMM-kp20             | 0.0376 | 24.97 | 0.160 | 1.222 | 0.118 |
> | SCORR+FOMM-kp20       | **0.0356** | **25.49** | **0.152** | **1.194** | **0.107** |
>
>
> ### Q2: User study, visualization of the CelebV-HQ dataset, and why didn't report the results of FNeVR on CelebV-HQ on Table 1
>
> * Thanks for the suggestion. We provide the user study in the **common response**.
> * Thanks for the suggestion. We have provided the anonymous link to video comparison results on the CelebV-HQ dataset, it can be accessed from the Area Chair. As can be seen in the provided video, on this more challenging dataset, our method generally performs the best in terms of learning finer motions, while other methods often produce blurry results, indicating that motion learned by their methods is blurrier than ours.
> * As claimed in lines 255-257 in the main paper, the authors of FNeVR didn't release their training codes and it didn't conduct experiments on this dataset. We had trouble in reproducing their paper due to the unclearness of the use of 3DMM parameters, so we didn't report their results on this dataset.

---

> ### Author Response · Authors · 2023-08-21
>
> Dear Reviewer 2aeH:
>
> Thank you for the recognition of our work. We addressed your concerns in the rebuttal. Could you please share your post comments? We are happy to discuss and address any remained concerns.
>
> Authors of the submission.

---

### Official Review · Reviewer_ELv3 · 2023-07-18

**Soundness:** 2 fair
**Presentation:** 3 good
**Contribution:** 2 fair
**Rating:** 4
**Confidence:** 4

**Summary:**

The paper presents a new approach to generate human face videos based on a source image and a driving video, which simultaneously learning both coarse and finer facial motions. The proposed approach utilizes a structure correlation volume constructed from keypoint features to provide non-prior-based motion information, which is used to iteratively refine the coarse motion flow that is estimated by a prior motion model.

The main contributions of the paper are:
1. A non-prior-based motion refinement approach to compensate for the inadequacy of existing prior-based motion models.
2. Utilize the keypoint features to build a structure correlation volume that represents the structure correspondence between the source and driving images across all spatial locations.
3. Extensive experiments on challenging benchmarks that demonstrate the effectiveness of the proposed approach in enhancing the capability of prior-based motion representation through learning the motion refinement.


**Strengths:**

1. The paper presents a novel approach to generate human face videos that simultaneously learns both coarse and finer facial motions. The proposed approach utilizes a structure correlation volume constructed from keypoint features to provide non-prior-based motion information, which is used to iteratively refine the coarse motion flow estimated by a prior motion model. The approach addresses a significant problem in the field of unsupervised face animation.

2. The paper is of good quality, and the method proposed is clear. The authors provide a detailed description of the approach, including the structure correlation volume and the motion refinement module. The experimental results demonstrate the effectiveness of the proposed approach in enhancing the capability of prior-based motion representation through learning the motion refinement.

3. The paper is well-written and easy to follow. The authors provide a clear and concise description of the proposed approach, including the key components and the experimental setup. The paper is well-organized, and the authors provide a clear summary of the contributions and limitations of the proposed approach.

4. The proposed approach has significant implications for the field of unsupervised face animation which addresses an important problem in this field: the inadequacy of existing prior-based motion models to capture detailed facial motions. The proposed approach is effective in enhancing the capability of prior-based motion representation through learning the motion refinement. The approach has potential applications in creating imaginative image animations for entertainment purposes, but it also has the potential to be used in creating deepfakes, which could have negative impacts. The authors acknowledge this limitation and provide recommendations for future work.


**Weaknesses:**

1.	The pictures in Figure 1 are too small, especially the optical flow map is not clear enough, no obvious changes can be seen before and after refinement. Also, is the schematic diagram of affine transformation in Figure 1 drawn based on a real example? Why does the affine transformation change so much after refinement?

2.	Straightforward combination of existing techniques. The innovation of this paper is not enough, the main innovation point lies in the motion refinement module. However, the method of correlation volume and iteratively refine optical flow used in it is very similar to the correlation matrix and iteratively update in some flow estimation[1,2,3] and correspondence estimation[4,5] methods, and this set of process of first constructing a 4D correlation volume, and then iteratively updating and optimizing optical flow has also been used in some neural style transfer(NST) methods[6,7], but the author did not clarify the differences between the module they used in this paper and related modules in these NST methods.

3.	There is an error in line 180 of the article: the author claims that the look up operation on the correlation volume is shown on the left side of Figure 2, but there is no corresponding content in Figure 2. Should Figure 2 be changed to Figure 3?

4.	In the comparison video provided in the supplemental material, the video of each method is too small, and it is difficult to see the tiny facial deformation details. It is recommended to arrange the videos of each method in the form of a nine-square grid, and enlarge the size of each video window.

5.	The experiments of verifying the effectiveness of the proposed method are insufficient. The framework used by your method is the framework of unsupervised image animation [8,9,10,11], which should be able to generate animation videos on any object category. In addition to human faces, this framework can also be applied to human bodies and animals. Previous unsupervised image animation methods [8,9,10,11] have also been tested on related datasets of different subjects, including TaiChiHD[9], TED-talks[10], and MGif[8]. Your method has only been tested on face-related datasets, but from your overall framework, I don't see any modules that restrict the method to only work on human face. Therefore, I think that the non-prior-based motion refinement module should be applied to datasets of different objects to further test its effectiveness.

6.	Lack of quantitative experiments on the Cross-identity task, such as user study that have been used in previous methods [8,9,10,11].

References

[1] Dosovitskiy, Alexey, Philipp Fischer, Eddy Ilg, Philip Hausser, Caner Hazirbas, Vladimir Golkov, Patrick van der Smagt, Daniel Cremers, and Thomas Brox. 2015. “FlowNet: Learning Optical Flow with Convolutional Networks.” In 2015 IEEE International Conference on Computer Vision (ICCV). doi:10.1109/iccv.2015.316.

[2] Teed, Zachary, and Jia Deng. 2020. “RAFT: Recurrent All-Pairs Field Transforms for Optical Flow.” In Computer Vision – ECCV 2020, Lecture Notes in Computer Science, 402–19. doi:10.1007/978-3-030-58536-5_24.

[3] Xu, Haofei, Jing Zhang, Jianfei Cai, Hamid Rezatofighi, and Dacheng Tao. 2022. “GMFlow: Learning Optical Flow via Global Matching.” In 2022 IEEE/CVF Conference on Computer Vision and Pattern Recognition (CVPR). doi:10.1109/cvpr52688.2022.00795.

[4] Kim, Seungryong, Stephen Lin, SangRyul Jeon, Dongbo Min, and Kwanghoon Sohn. 2018. “Recurrent Transformer Networks for Semantic Correspondence.” arXiv: Computer Vision and Pattern Recognition,arXiv: Computer Vision and Pattern Recognition, October.

[5] Zhang, Pan, Bo Zhang, Dong Chen, Lu Yuan, and Fang Wen. 2020. “Cross-Domain Correspondence Learning for Exemplar-Based Image Translation.” In 2020 IEEE/CVF Conference on Computer Vision and Pattern Recognition (CVPR). doi:10.1109/cvpr42600.2020.00519.

[6] Liu, Xiaochang, Xuanyi Li, Ming-Ming Cheng, and Peter Hall. 2020. “Geometric Style Transfer.” Cornell University - arXiv,Cornell University - arXiv, July.

[7] Liu, Xiao-Chang, Yong-Liang Yang, and Peter Hall. 2021. “Learning to Warp for Style Transfer.” In 2021 IEEE/CVF Conference on Computer Vision and Pattern Recognition (CVPR). doi:10.1109/cvpr46437.2021.00370.

[8] Siarohin, Aliaksandr, Stephane Lathuiliere, Sergey Tulyakov, Elisa Ricci, and Nicu Sebe. 2019. “Animating Arbitrary Objects via Deep Motion Transfer.” In 2019 IEEE/CVF Conference on Computer Vision and Pattern Recognition (CVPR). doi:10.1109/cvpr.2019.00248.

[9] Siarohin, Aliaksandr, Stéphane Lathuilière, Sergey Tulyakov, Elisa Ricci, and Nicu Sebe. 2019. “First Order Motion Model for Image Animation.” Neural Information Processing Systems,Neural Information Processing Systems, January.

[10] Siarohin, Aliaksandr, Oliver J. Woodford, Jian Ren, Menglei Chai, and Sergey Tulyakov. 2021. “Motion Representations for Articulated Animation.” In 2021 IEEE/CVF Conference on Computer Vision and Pattern Recognition (CVPR). doi:10.1109/cvpr46437.2021.01344.

[11] Zhao, Jian, and Hui Zhang. n.d. “Thin-Plate Spline Motion Model for Image Animation.”


**Questions:**

It’s suggested that the author should clarify the differences between the module they used in this paper and the related modules in some NST methods [6,7]. Also, experiments on more datasets of different object categories are needed.

**Limitations:**

Yes, they have adequately addressed the limitations.

---

> ### Author Rebuttal · Authors · 2023-08-09
>
> ## Response to Reviewer ELv3
>
> ### Q1: Presentation of Fig.1, cross reference of Fig.3 (line 180), composing type of supplementary videos.
>
> * Sorry for the confusion. The affine transformation is for illustration purposes, not from real examples. But the two motion flows are influenced by a pair of real source and driving images. The motion flow is actually obtained by compositing a set of local affine transformations in FOMM, while we use a single global affine transformation for the purpose of the simple introduction. For better understanding,  we provide the visualization of refined motion flows across all iterations in the video (the last few seconds). We will improve Fig.1 in the later version.
> * You are right, Figure 2 should be changed to Figure 3, we will correct this in the later version.
> * Thanks for the suggestion! We will rearrange the videos of each method to form a nine-square result for better comparison. Benefiting from your suggestion, our newly updated video results are clearer than before.
>
>
> ### Q2: The innovation of this paper, difference with optical flow methods and neural style transfer methods.
>
> * Thank you for the valuable thoughts. We addressed these concerns in the **common response**. And we sincerely urge your reconsideration on this point.
>
>
> ### Q3: The experiments of verifying the effectiveness of the proposed method are insufficient, other object animation experiments are needed.
>
> Similar to DaGAN and FNeVR which explored facial specific information such as facial depth and 3DMM information, for helping or enhancing learning the local affine motion representation of faces, we would like to emphasize that our method is better applied to face animation than human body animation. The reason is that the locally rigid assumption is reasonable and robust for human bodies, and thus our non-prior based motion refinement approach may bring no significant improvements. However, this assumption can be easily violated for the human face, since non-rigid deformation is very normal in a local face area, and this special attribute of face motion motivates us to investigate the non-prior based motion refinement. Existing methods such as FOMM, MRAA, DAM and TPSM generally focused on finding better parametric motion models, while bringing considerable improvements on the human body animation, it ignores the importance of non-prior motion for face animation. Therefore, our method is more specially designed for face animation, and we believe the experiments on face animation are sufficient to validate its effectiveness.
>
> As a thought, our method may help the deformation modeling like clothes of the human body, which may need the finer motion modeling compared to the body itself. However, currently the more urgent problem in body animation is to more accurately find the body structures and then it can be better transferred.
>
> We here simply conduct additional experiments on the TEDTalks dataset. As it shows, our method can improve the earlier baseline FOMM by a considerable margin, however, it didn't improve the sota baseline TPSM much especially for the motion related metric AKD.
>
>
> |            | L1         |    AKD    |   AED |
> | :--------: | :--------: | :--------:| :----:|
> | FOMM       | 0.0281     | 4.413     | 0.129 |
> | SCORR+FOMM | 0.0269     | 3.535     | 0.119 |
> | TPSM       | 0.0271     | 3.501     | 0.126 |
> | SCORR+TPSM | 0.0265     | 3.499     | 0.118 |
>
>
>
> ### Q4: Lack of quantitative experiments on the Cross-identity task, such as user study that have been used in previous methods.
>
> * Thanks for the suggestion. Initially, we have conducted a cross-identity task in the supplementary, evaluating the pose and expression quality of generated videos. And we here further present a comparison on the CSIM and FID metrics. We addressed this concern in the **common response**.

---

> ### Author Response · Authors · 2023-08-19
> **On Responses of the Rebuttal**
>
> Dear Reviewer ELv3:
>
> We addressed your concerns in the rebuttal on innovation, experiments, and presentation. We sincerely urge your reconsideration of this paper and hope that you could share your post comments on our rebuttal. Thank you very much.
>
> Authors of the submission.

---

### Author Rebuttal · Authors · 2023-08-09

## Common Response

We thank all Reviewers for the valuable feedback and insightful comments. We appreciate the reviewer's positive comments regarding the novelty and the effectiveness of our method. We now clarify the reviewer's common questions as follows.

### Q1: Difference with optical flow and neural style transfer methods (Reviewer ELv3 and pDRj).

* We should emphasize that a key difference between our method and optical flow methods lies in building the correlation volume, which we discussed in the related work from line 90 to line 104 in the main paper. Optical flow methods compute the **appearance feature correspondence**  of two consecutive frames, while our key idea is to learn the **structural correspondence**. The methods in optical flow can't be directly applied to face animation, as computing appearance feature similarity between driving and source images will naively leak the driving appearance, which is not expected in face animation. As a result, the optical flow methods will directly find appearance from the source face that is similar to the driving face. However, the task of face animation aims to find motion between source and driving (structure correspondence), but not to find appearance correspondence.
Existing face animation methods generally model the structural correspondence by local parametric transformations between the source and driving frame, which ignores the importance of **building the non-prior based correspondence**. Thus our efforts in constructing the structure correlation volume, though inspired by the optical flow method RAFT, are meaningful and motivatable for face animation. And based on this motivation, we made the first attempt to find a way to introduce the non-prior evidence for motion refinement. Therefore, we believe our method is novel and can fill some gaps in the field of face animation.
* Thank you for suggesting the neural style transfer methods. And we have carefully read the paper [1-2] in this area. We summarize two main differences between our method and theirs, the way to build correspondence and the purpose/motivation of building it. On one hand, they still lie in a similar paradigm with optical flow methods, which computes the visual appearance similarity between the content image and the geometric style image. While our method uses the structure features as input to build correlation volume. More essentially, on the other hand, the neural style transfer methods [1-2] utilize the correspondence matrix to estimate **a global parametric transformation**, which is totally different from our motivation that aims to use the structure correlation for estimating the **non-parametric motion refinement** across all spatial locations. This non-parametric motion modeling is specifically designed for face animation. And it is generally ignored by existing methods. Thanks for suggesting the paper again. We will include these discussions together with the optical flow methods in the later version.

[1] Geometric Style Transfer. arXiv 2020.
[2] Learning to Warp for Style Transfer. CVPR2021

### Q2: Run time and memory evaluation (Reviewer mUhK and zuqN)

Thank reviewers for the suggestion. We here present the inference-stage run time evaluation. ALL results are obtained by running reconstruction experiments on a single NVIDIA 3090 GPU. Compared to year-2022 methods, the memory cost is similar among all methods, the FPS of our method is slightly worse than that of LIA, DAM, TPSM, and FNeVR, but better than DaGAN. Considering that our method produces the best animation quality, our method achieves better trade-off between performance and inference speed.
|  | FOMM | MRAA  | LIA | DAM | DaGAN | TPSM | FNeVR | Ours |
|:--:|:------:|:-------:|:-----:|:-----:|:-------:|:------:|:-------:|:------:|
| Memory (G)$\downarrow$  | 6.48 | 6.51  | **2.63** | 6.49  | 7.07  | 6.82  | 6.53  | 7.02  |
| FLOPs (G)$\downarrow$   | **56.06** | 61.07 | 88.32 | 56.39 | 89.78 | 142.4 | 130.2 | 116.5 |
| FPS$\uparrow$        | **41.64** | 33.04 | 20.98 | 30.29 | 17.72 | 19.96 | 21.83 | 18.57 |
### Q3: Cross-identity evaluation (Reviewer ELv3, 2aeH) and identity preservation (Reviewe mUhK, SbHP and zuqN )
* Thank reviewers for the suggestion. Initially we compare the ARD and AUH methric for cross-identity animation in the Table. A1 of the supplementary, which evaluates the pose and expression quality of generatedimages . We here further compare the CSIM (cosine similarity of the face embedding using arcface[1]) and FID metric. As seen in the following table, our method also performs best in terms of CSIM and FID, indicating that our method preserves better the source identity information. It should be noted that, in self reconstruction task, the AED metric is also designed for evaluating identity preservation, and our method generally performs best in term of this metric.
| | FOMM  | MRAA  | LIA   | DAM   | DaGAN | TPSM  | FNeVR | Ours           |
|:--:|:------:|:-------:|:-----:|:-----:|:-------:|:------:|:-------:|:------:|
| ARD$\downarrow$        | 3.122 | 2.678 | 3.883 | 2.669 | 3.090 | 2.724 | 2.755 | **2.399** |
| AUH$\downarrow$                   | 0.850 | 0.729 | 0.772 | 0.717 | 0.751 | 0.668 | 0.751 | **0.625** |
| CSIM$\uparrow$        | 0.702 | 0.678 | 0.706 | 0.698 | 0.682 | 0.689 | 0.714 | **0.722** |
| FID$\downarrow$                   | 70.00 | 69.29 | 71.01 | 69.16 | 68.50 | 68.67 | 68.42 | **66.27** |

[1] ArcFace: Additive Angular Margin Loss for Deep Face Recognition, CVPR2019
* As suggested, we also conducted a user study experiment. Specifically, 20 participants are asked to evaluate 50 randomly generated videos of different methods, according to the transferred motion and the preserved identity. There are a total of 1000 votes, and we compute the ratio in favor of our method. It can be seen that our method is generally more favorable compared to TPSM and FNeVR.
| Ours $vs$ TPSM | Ours $vs$ FNeVR  |
|:------:|:-------:|
| 61.1% | 56.9%  |

---

### Author Response · Authors · 2023-08-10
**Anonymous link of the provided video.**

https://user-images.githubusercontent.com/38600167/259588792-0b7d0816-e316-4304-8d50-dd1a54ddc48d.mp4

---

### Decision · Program_Chairs · 2023-09-21

**Decision:**

Accept (poster)

**Comment:**

Solid paper with good results, it improves the state of the art. The ML contribution is not the strongest, as the technical improvements are relatively straightforward but at the same time they are necessary for accurate facial animation.